# On the Complexity of Learning Neural Networks

**Le Song**
Georgia Institute of Technology
Atlanta, GA 30332
lsong@cc.gatech.edu

**Santosh Vempala**
Georgia Institute of Technology
Atlanta, GA 30332
vempala@gatech.edu

**John Wilmes**
Georgia Institute of Technology
Atlanta, GA 30332
wilmesj@gatech.edu

**Bo Xie**
Georgia Institute of Technology
Atlanta, GA 30332
bo.xie@gatech.edu

## Abstract

The stunning empirical successes of neural networks currently lack rigorous theoretical explanation. What form would such an explanation take, in the face of existing complexity-theoretic lower bounds? A first step might be to show that data generated by neural networks with a single hidden layer, smooth activation functions and benign input distributions can be learned efficiently. We demonstrate here a comprehensive lower bound ruling out this possibility: for a wide class of activation functions (including all currently used), and inputs drawn from any logconcave distribution, there is a family of one-hidden-layer functions whose output is a sum gate, that are hard to learn in a precise sense: any *statistical query* algorithm (which includes all known variants of stochastic gradient descent with any loss function) needs an exponential number of queries even using tolerance inversely proportional to the input dimensionality. Moreover, this hard family of functions is realizable with a small (sublinear in dimension) number of activation units in the single hidden layer. The lower bound is also robust to small perturbations of the true weights. Systematic experiments illustrate a phase transition in the training error as predicted by the analysis.

## 1   Introduction

It is well-known that Neural Networks (NN's) provide universal approximate representations [11, 6, 2] and under mild assumptions, i.e., any real-valued function can be approximated by a NN. This holds for a wide class of activation functions (hidden layer units) and even with only a single hidden layer (although there is a trade-off between depth and width [8, 20]). Typically learning a NN is done by stochastic gradient descent applied to a loss function comparing the network's current output to the values of the given training data; for regression, typically the function is just the least-squares error. Variants of gradient descent include drop-out, regularization, perturbation, batch gradient descent etc. In all cases, the training algorithm has the following form:

Repeat:

1. Compute a fixed function $F_W(.)$ defined by the current network weights $W$ on a subset of training examples.

2. Use $F_W(.)$ to update the current weights $W$.

The empirical success of this approach raises the question: what can NN's learn efficiently *in theory*? In spite of much effort, at the moment there are no satisfactory answers to this question, even with reasonable assumptions on the function being learned and the input distribution.

When learning involves some computationally intractable optimization problem, e.g., learning an intersection of halfspaces over the uniform distribution on the Boolean hypercube, then any training algorithm is unlikely to be efficient. This is the case even for improper learning (when the complexity of the hypothesis class being used to learn can be greater than the target class). Such lower bounds are unsatisfactory to the extent they rely on discrete (or at least nonsmooth) functions and distributions. What if we assume that the function to be learned is generated by a NN with a single hidden layer of smooth activation units, and the input distribution is benign? Can such functions be learned efficiently by gradient descent?

Our main result is a lower bound, showing a simple and natural family of functions generated by 1-hidden layer NN's using any known activation function (e.g., sigmoid, ReLU), with each input drawn from a logconcave input distribution (e.g., Gaussian, uniform in an interval), are hard to learn by a wide class of algorithms, including those in the general form above. Our finding implies that efficient NN training algorithms need to use stronger assumptions on the target function and input distribution, more so than Lipschitzness and smoothness even when the true data is generated by a NN with a single hidden layer.

The idea of the lower bound has two parts. First, NN updates can be viewed as *statistical queries* to the input distribution. Second, there are many very different 1-layer networks, and in order to learn the correct one, any algorithm that makes only statistical queries of not too small accuracy has to make an exponential number of queries. The lower bound uses the SQ framework of Kearns [13] as generalized by Feldman et al. [9].

## 1.1 Statistical query algorithms

A statistical query (SQ) algorithm is one that solves a computational problem over an input distribution; its interaction with the input is limited to querying the expected value of of a bounded function up to a desired accuracy. More precisely, for any integer $t > 0$ and distribution $D$ over $X$, a VSTAT($t$) **oracle** takes as input a **query function** $f : X \to [0, 1]$ with expectation $p = \mathbb{E}_D(f(x))$ and returns a value $v$ such that

$$\left| \mathbb{E}_{x \sim D}(f(x)) - v \right| \leq \max\left\{ \frac{1}{t}, \sqrt{\frac{p(1-p)}{t}} \right\}.$$

The bound on the RHS is the standard deviation of $t$ independent Bernoulli coins with desired expectation, i.e., the error that even a random sample of size $t$ would yield. In this paper, we study SQ algorithms that access the input distribution only via the VSTAT($t$) oracle. The remaining computation is unrestricted and can use randomization (e.g., to determine which query to ask next).

In the case of an algorithm training a neural network via gradient descent, the relevant query functions are derivatives of the loss function.

The statistical query framework was first introduced by Kearns for supervised learning problems [14] using the STAT($\tau$) oracle, which, for $\tau \in \mathbb{R}_+$, responds to a query function $f : X \to [0, 1]$ with a value $v$ such that $|\mathbb{E}_D(f) - v| \leq \tau$. The STAT($\sqrt{\tau}$) oracle can be simulated by the VSTAT($O(1/\tau)$) oracle. The VSTAT oracle was introduced by [9] who extended these oracles to general problems over distributions.

## 1.2 Main result

We will describe a family $\mathcal{C}$ of functions $f : \mathbb{R}^n \to \mathbb{R}$ that can be computed exactly by a small NN, but cannot be efficiently learned by an SQ algorithm. While our result applies to all commonly used activation units, we will use sigmoids as a running example. Let $\sigma(z)$ be the sigmoid gate that goes to 0 for $z < 0$ and goes to 1 for $z > 0$. The sigmoid gates have sharpness parameter $s$, i.e., $\sigma(x) = \sigma_s(x) = (1 + e^{-sx})^{-1}$. Note that the parameter $s$ also bounds the Lipschitz constant of $\sigma(x)$.

A function $f : \mathbb{R}^n \to \mathbb{R}$ can be computed exactly by a single layer NN with sigmoid gates precisely when it is of the form $f(x) = h(\sigma(g(x)))$, where $g : \mathbb{R}^n \to \mathbb{R}^m$ and $h : \mathbb{R}^m \to \mathbb{R}$ are affine, and $\sigma$ acts component-wise. Here, $m$ is the number of hidden units, or sigmoid gates, of the of the NN.

In the case of a learning problem for a class $\mathcal{C}$ of functions $f : X \to \mathbb{R}$, the input distribution to the algorithm is over labeled examples $(x, f^*(x))$, where $x \sim D$ for some underlying distribution $D$ on $X$, and $f^* \in \mathcal{C}$ is a fixed concept (function).

As mentioned in the introduction, we can view a typical NN learning algorithm as a statistical query (SQ) algorithm: in each iteration, the algorithm constructs a function based on its current weights (typically a gradient or subgradient), evaluates it on a *batch* of random examples from the input distribution, then uses the evaluations to update the weights of the NN. Then we have the following result.

**Theorem 1.1.** *Let $n \in \mathbb{N}$, and let $\lambda, s \geq 1$. There exists an explicit family $\mathcal{C}$ of functions $f : \mathbb{R}^n \to [-1, 1]$, representable as a single hidden layer neural network with $O(s\sqrt{n} \log(\lambda sn))$ sigmoid units of sharpness $s$, a single output sum gate and a weight matrix with condition number $O(\mathrm{poly}(n, s, \lambda))$, and an integer $t = \Omega(s^2 n)$ s.t. the following holds. Any (randomized) SQ algorithm $A$ that uses $\lambda$-Lipschitz queries to $\mathrm{VSTAT}(t)$ and weakly learns $\mathcal{C}$ with probability at least $1/2$, to within regression error $1/\sqrt{t}$ less than any constant function over i.i.d. inputs from any logconcave distribution of unit variance on $\mathbb{R}$ requires $2^{\Omega(n)}/(\lambda s^2)$ queries.*

The Lipschitz assumption on the statistical queries is satisfied by all commonly used algorithms for training neural networks can be simulated with Lipschitz queries (e.g., gradients of natural loss functions with regularizers). This assumption can be omitted if the output of the hard-to-learn family $\mathcal{C}$ is represented with bounded precision.

Informally, Theorem 1.1 shows that there exist simple realizable functions that are not efficiently learnable by NN training algorithms with polynomial batch sizes, assuming the algorithm allows for error as much as the standard deviation of random samples for each query. We remark that in practice, large batch sizes are seldom used for training NNs, not just for efficiency, but also since moderately noisy gradient estimates are believed to be useful for avoiding bad local minima. Even NN training algorithms with larger batch sizes will require $\Omega(t)$ samples to achieve lower error, whereas the NNs that represent functions in our class $\mathcal{C}$ have only $\widetilde{O}(\sqrt{t})$ parameters.

Our lower bound extends to a broad family of activation units, including all the well-known ones (ReLU, sigmoid, softplus etc., see Section 3.1). In the case of sigmoid gates, the functions of $\mathcal{C}$ take the following form (cf. Figure 1.1). For a set $S \subseteq \{1, \ldots, n\}$, we define $f_{m,S}(x_1, \ldots, x_n) = \phi_m(\sum_{i \in S} x_i)$, where

$$\phi_m(x) = -(2m + 1) + \sum_{k=-m}^{m} \sigma\left(x - \frac{(4k - 1)}{s}\right) + \sigma\left(\frac{(4k + 1)}{s} - x\right). \qquad (1.1)$$

Then $\mathcal{C} = \{f_{m,S} : S \subseteq \{1, \ldots, n\}\}$. We call the functions $f_{m,S}$, along with $\phi_m$, the *s-wave* functions. It is easy to see that they are smooth and bounded. Furthermore, the size of the NN representing this hard-to-learn family of functions is only $\tilde{O}(s\sqrt{n})$, assuming the query functions (e.g., gradients of loss function) are $\mathrm{poly}(s, n)$-Lipschitz. We note that the lower bounds hold regardless of the architecture of the model, i.e., NN used to learn.

Our lower bounds are asymptotic, but we show empirically in Section 4 that they apply even at practical values of $n$ and $s$. We experimentally observe a threshold for the quantity $s\sqrt{n}$, above which stochastic gradient descent fails to train the NN to low error—that is, regression error below that of the best constant approximation— regardless of choices of gates, architecture used to learning, learning rate, batch size, etc.

The condition number upper bound for $\mathcal{C}$ is significant in part because there do exist SQ algorithms for learning certain families of simple NNs with time complexity polynomial in the condition number of the weight matrix (the tensor factorization based algorithm of Janzamin et al. [12] can easily be seen to be SQ). Our results imply that this dependence cannot be substantially improved (see Section 1.3).

**Remark 1.** The class of input distributions can be relaxed further. Rather than being a product distribution, it suffices if the distribution is in isotropic position and invariant under reflections across

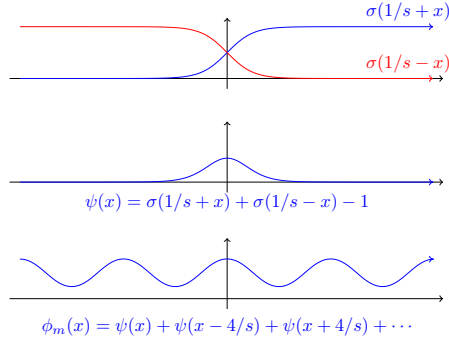

Figure 1.1: (a) The sigmoid function, the $L^1$-function $\psi$ constructed from sigmoid functions, and the nearly-periodic "wave" function $\phi$ constructed from $\psi$. (b) The architecture of the NNs computing the wave functions.

and permutations of coordinate axes. And instead of being logconcave, it suffices for marginals to be unimodal with variance $\sigma$, density $O(1/\sigma)$ at the mode, and density $\Omega(1/\sigma)$ within a standard deviation of the mode.

Overall, our lower bounds suggest that even the combination of small network size, smooth, standard activation functions, and benign input distributions is insufficient to make learning a NN easy, even improperly via a very general family of algorithms. Instead, stronger structural assumptions on the NN, such as a small condition number, and very strong structural properties on the input distribution, are necessary to make learning tractable. It is our hope that these insights will guide the discovery of provable efficiency guarantees.

## 1.3   Related Work

There is much work on complexity-theoretic hardness of learning neural networks [4, 7, 15]. These results have shown the hardness of learning functions representable as small (depth 2) neural networks over discrete input distributions. Since these input distributions bear little resemblance to the real-world data sets on which NNs have seen great recent empirical success, it is natural to wonder whether more realistic distributional assumptions might make learning NNs tractable. Our results suggest that benign input distributions are insufficient, even for functions realized as small networks with standard, smooth activation units.

Recent independent work of Shamir [17] shows a smooth family of functions for which the gradient of the squared loss function is not informative for training a NN over a Gaussian input distribution (more generally, for distributions with rapidly decaying Fourier coefficients). In fact, for this setting the paper shows an exponentially small bound on the gradient, relying on the fine structure of the Gaussian distribution and of the smooth functions (see [16] for a follow-up with experiments and further ideas). These smooth functions cannot be realized in small NNs using the most commonly studied activation units (though a related non-smooth family of functions for which the bounds apply can be realized by larger NNs using ReLU units). In contrast our bounds are (a) in the more general SQ framework, and in particular apply regardless of the loss function, regularization scheme, or specific variant of gradient descent (b) apply to functions actually realized as small NNs using any of a wide family of activation units (c) apply to any logconcave input distribution and (d) are robust to small perturbations of the input layer weights.

Also related is the tensor-based algorithm of Janzamin et al. [12] to learn a 1-layer network under nondegeneracy assumptions on the weight matrix. The complexity is polynomial in the dimension, size of network being learned and condition number of the weight matrix. Since their tensor decomposition can also be implemented as a statistical query algorithm, our results give a lower bound indicating that such a polynomial dependence on the dimension and condition number is unavoidable.

Other algorithmic results for learning NNs apply in very restricted settings. For example, polynomial-time bounds are known for learning NNs with a single hidden ReLU layer over Gaussian inputs under

the assumption that the hidden units use disjoint sets of inputs [5], as well as for learning a single ReLU [10] and for learning sparse polynomials via NNs [1].

## 1.4 Proof ideas

To prove Theorem 1.1, we wish to estimate the number of queries used by a statistical query algorithm learning the family of $s$-wave functions, regardless of the strategy employed by the algorithm. To that end, we estimate the *statistical dimension* of the family of $s$-wave functions. Statistical dimension is a key concept in the study of SQ algorithms, and is known to characterize the query complexity of supervised learning via SQ algorithms [3, 19, 9]. Briefly, a family $\mathcal{C}$ of distributions (e.g., over labeled examples) has "statistical dimension $d$ with average correlation $\bar{\gamma}$" if every $(1/d)$-fraction of $\mathcal{C}$ has average correlation $\bar{\gamma}$; this condition implies that $\mathcal{C}$ cannot be learned with fewer than $O(d)$ queries to $\mathrm{VSTAT}(O(1/\bar{\gamma}))$. See Section 2 for precise statements.

The SQ literature for supervised learning of boolean functions is rich. However, lower bounds for regression problems in the SQ framework have so far not appeared in the literature, and the existing notions of statistical dimension are too weak for this setting. We state a new, strengthened notion of statistical dimension for regression problems (Definition 2), and show that lower bounds for this dimension transfer to query complexity bounds (Theorem 2.1). The essential difference from the statistical dimension for learning is that we must additionally bound the average covariances of *indicator functions* (or, rather, continuous analogues of indicators) on the outputs of functions in $\mathcal{C}$. The essential claim in our lower bounds is therefore in showing that a typical pair of (indicator functions on outputs of) $s$-wave functions has small covariance.

In other words, to prove Theorem 1.1, it suffices to upper-bound the quantity

$$\mathbb{E}[(\chi \circ f_{m,S})(\chi \circ f_{m,T})] - \mathbb{E}[\chi \circ f_{m,S}]\mathbb{E}[\chi \circ f_{m,T}] \tag{1.2}$$

for most pairs $f_{m,S}, f_{m,T}$ of $s$-wave functions, where $\chi$ is some smoothed version of an indicator function. Write $h(t) = \chi(\phi_m(t))$, so $\chi(f_{m,S}(x_1, \ldots, x_n)) = h(\sum_{i \in S} x_i)$. We have

$$\mathbb{E}_{(x_1,\ldots,x_n) \sim D}(h(\sum_{i \in S} x_i) h(\sum_{i \in T} x_i) \mid \sum_{i \in S \cap T} x_i = z)$$

$$= \mathbb{E}_{x_i, i \in S \setminus T}(h(\sum_{i \in S \setminus T} x_i + z)) \mathbb{E}_{x_i, i \in T \setminus S}(h(\sum_{i \in T \setminus S} x_i + z)).$$

So to estimate Eq. (1.2), it suffices to show that the expectation of $h(\sum_{i \in S} x_i)$ doesn't change much when we condition on the value of $z = \sum_{i \in S \cap T} x_i$.

We now observe that if $\chi$ is Lipschitz, and $\phi_m$ is "close to" a periodic function with period $\theta > 0$, then $h$ is also "close to" a periodic function with period $\theta > 0$ (see Section 3 for a precise statement). Under this near-periodicity assumption, we are now able to show for any logconcave distribution $D'$ on $\mathbb{R}$ of variance $\sigma > \theta$, and any translation $z \in \mathbb{R}$, that

$$\left| \mathbb{E}_{x \sim D}(h(x + z) - h(x)) \right| = O\left(\frac{\theta}{\sigma}\right) \mathbb{E}_{x \sim D}(|h(x)|).$$

In particular, conditioning on the value of $z = \sum_{i \in S \cap T} x_i$ has little effect on the value of $h(\sum_{i \in S} x_i)$. The combination of these observations gives the query complexity lower bound. Precise statements of some of the technical lemmas are given in Section 3; the complete proof appears in the full version of this paper [18].

## 2 Statistical dimension

We now give a precise definition of the *statistical dimension with average correlation* for regression problems, extending the concept introduced in [9].

Let $\mathcal{C}$ be a finite family of functions $f : X \to \mathbb{R}$ over some domain $X$, and let $D$ be a distribution over $X$. The average covariance and the average correlation of $\mathcal{C}$ with respect to $D$ are

$$\mathrm{Cov}_D(\mathcal{C}) = \frac{1}{|\mathcal{C}|^2} \sum_{f,g \in \mathcal{C}} \mathrm{Cov}_D(f,g) \quad \text{and} \quad \rho_D(\mathcal{C}) = \frac{1}{|\mathcal{C}|^2} \sum_{f,g \in \mathcal{C}} \rho_D(f,g)$$

where $\rho_D(f, g) = \mathrm{Cov}_D(f, g)/\sqrt{\mathrm{Var}(f)\,\mathrm{Var}(g)}$ when both $\mathrm{Var}(f)$ and $\mathrm{Var}(g)$ are nonzero, and $\rho_D(f, g) = 0$ otherwise.

For $y \in \mathbb{R}$ and $\epsilon > 0$, we define the $\epsilon$-**soft indicator function** $\chi_y^{(\epsilon)} : \mathbb{R} \to \mathbb{R}$ as

$$\chi_y^{(\epsilon)}(x) = \chi_y(x) = \max\{0, 1/\epsilon - (1/\epsilon)^2 |x - y|\}.$$

So $\chi_y$ is $(1/\epsilon)^2$-Lipschitz, is supported on $(y - \epsilon, y + \epsilon)$, and has norm $\|\chi_y\|_1 = 1$.

**Definition 2.** Let $\bar{\gamma} > 0$, let $D$ be a probability distribution over some domain $X$, and let $\mathcal{C}$ be a family of functions $f : X \to [-1, 1]$ that are identically distributed as random variables over $D$. The **statistical dimension** of $\mathcal{C}$ relative to $D$ with average covariance $\bar{\gamma}$ and precision $\epsilon$, denoted by $\epsilon\text{-SDA}(\mathcal{C}, D, \bar{\gamma})$, is defined to be the largest integer $d$ such that the following holds: for every $y \in \mathbb{R}$ and every subset $\mathcal{C}' \subseteq \mathcal{C}$ of size $|\mathcal{C}'| > |\mathcal{C}|/d$, we have $\rho_D(\mathcal{C}') \leq \bar{\gamma}$. Moreover, $\mathrm{Cov}_D(\mathcal{C}'_y) \leq (\max\{\epsilon, \mu(y)\})^2 \bar{\gamma}$ where $\mathcal{C}'_y = \{\chi_y^{(\epsilon)} \circ f : f \in \mathcal{C}\}$ and $\mu(y) = E_D(\chi_y^{(\epsilon)} \circ f)$ for some $f \in \mathcal{C}$.

Note that the parameter $\mu(y)$ is independent of the choice of $f \in \mathcal{C}$. The application of this notion of dimension is given by the following theorem.

**Theorem 2.1.** *Let $D$ be a distribution on a domain $X$ and let $\mathcal{C}$ be a family of functions $f : X \to [-1, 1]$ identically distributed as random variables over $D$. Suppose there is $d \in \mathbb{R}$ and $\lambda \geq 1 \geq \bar{\gamma} > 0$ such that $\epsilon\text{-SDA}(\mathcal{C}, D, \bar{\gamma}) \geq d$, where $\epsilon \leq \bar{\gamma}/(2\lambda)$. Let $A$ be a randomized algorithm learning $\mathcal{C}$ over $D$ with probability greater than $1/2$ to regression error less than $\Omega(1) - 2\sqrt{\bar{\gamma}}$. If $A$ only uses queries to $\mathrm{VSTAT}(t)$ for some $t = O(1/\bar{\gamma})$, which are $\lambda$-Lipschitz at any fixed $x \in X$, then $A$ uses $\Omega(d)$ queries.*

A version of the theorem for Boolean functions is proved in [9]. For completeness, in the full version of this paper [18] we include a proof of Theorem 2.1, following ideas in [19, Theorem 2].

As a consequence of Theorem 2.1, there is no need to consider an SQ algorithm's query strategy in order to obtain lower bounds on its query complexity. Instead, the lower bounds follow directly from properties of the concept class itself, in particular from bounds on average covariances of indicator functions. Theorem 1.1 will therefore follow from Theorem 2.1 by analyzing the statistical dimension of the $s$-wave functions.

# 3 Estimates of statistical dimension for one-layer functions

We now present the most general context in which we obtain SQ lower bounds.

A function $\phi : \mathbb{R} \to \mathbb{R}$ is $(M, \delta, \theta)$-**quasiperiodic** if there exists a function $\tilde{\phi} : \mathbb{R} \to \mathbb{R}$ which is periodic with period $\theta$ such that $|\phi(x) - \tilde{\phi}(x)| < \delta$ for all $x \in [-M, M]$. In particular, any periodic function with period $\theta$ is $(M, \delta, \theta)$-quasiperiodic for all $M, \delta > 0$.

**Lemma 3.1.** *Let $n \in \mathbb{N}$ and let $\theta > 0$. There exists $\bar{\gamma} = O(\theta^2/n)$ such that for all $\epsilon > 0$, there exist $M = O(\sqrt{n} \log(n/(\epsilon\theta)))$ and $\delta = \Omega(\epsilon^3 \theta/\sqrt{n})$ and a family $\mathcal{C}_0$ of affine functions $g : \mathbb{R}^n \to \mathbb{R}$ of bounded operator norm with the following property. Suppose $\phi : \mathbb{R} \to [-1, 1]$ is $(M, \delta, \theta)$-quasiperiodic and $\mathrm{Var}_{x \sim U(0,\theta)}(\phi(x)) = \Omega(1)$. Let $D$ be logconcave distribution with unit variance on $\mathbb{R}$. Then for $\mathcal{C} = \{\phi \circ g : g \in \mathcal{C}_0\}$, we have $\epsilon\text{-SDA}(\mathcal{C}, D^n, \bar{\gamma}) \geq 2^{\Omega(n)} \epsilon \theta^2$. Furthermore, the functions of $\mathcal{C}$ are identically distributed as random variables over $D^n$.*

In other words, we have statistical dimension bounds (and hence query complexity bounds) for functions that are sufficiently close to periodic. However, the activation units of interest are generally monotonic increasing functions such as sigmoids and ReLUs that are quite far from periodic. Hence, in order to apply Lemma 3.1 in our context, we must show that the activation units of interest can be combined to make nearly periodic functions.

As an intermediate step, we analyze activation functions in $L^1(\mathbb{R})$, i.e., functions whose absolute value has bounded integral over the whole real line. These $L^1$-functions analyzed in our framework are themselves constructed as affine combinations of the usual activation functions. For example, for the sigmoid unit with sharpness $s$, we study the following $L^1$-function (cf. (1.1)):

$$\psi(x) = \sigma\left(\frac{1}{s} + x\right) + \sigma\left(\frac{1}{s} - x\right) - 1. \tag{3.1}$$

We now describe the properties of the integrable functions $\psi$ that will be used in the proof.

**Definition 3.** For $\psi \in L^1(\mathbb{R})$, we say the **essential radius** of $\psi$ is the number $r \in \mathbb{R}$ such that $\int_{-r}^{r} |\psi| = (5/6)\|\psi\|_1$.

**Definition 4.** We say $\psi \in L^1(\mathbb{R})$ has the **mean bound property** if for all $x \in \mathbb{R}$ and $\epsilon > 0$, we have

$$\psi(x) = O\left(\frac{1}{\epsilon} \int_{x-\epsilon}^{x+\epsilon} |\psi(x)|\right).$$

In particular, if $\psi$ is bounded, and monotonic nonincreasing (resp. nondecreasing) for sufficiently large positive (resp. negative) inputs, then $\psi$ satisfies Definition 4. Alternatively, it suffices for $\psi$ to have bounded first derivative.

To complete the proof of Theorem 1.1, we show that we can combine activation units $\psi$ satisfying the above properties in a function which is close to periodic, i.e., which satisfies the hypotheses of Lemma 3.1 above.

**Lemma 3.2.** *Let $\psi \in L^1(\mathbb{R})$ have the mean bound property and let $r > 0$ be such that $\psi$ has essential radius at most $r$ and $\|\psi\|_1 = \Theta(r)$. Let $M, \delta > 0$. Then there is a pair of affine functions $h : \mathbb{R}^m \to \mathbb{R}$ and $g : \mathbb{R} \to \mathbb{R}^m$ such that if $\phi(x) = h(\psi(g(x)))$, where $\psi$ is applied component-wise, then $\phi$ is $(M, \delta, 4r)$-quasiperiodic. Furthermore, $\phi(x) \in [-1, 1]$ for all $x \in \mathbb{R}$, and $\mathrm{Var}_{x \sim U(0,4r)}(\phi(x)) = \Omega(1)$, and we may take $m = (1/r) \cdot O(\max\{m_1, M\})$, where $m_1$ satisfies*

$$\int_{m_1}^{\infty} (|\psi(x)| + |\psi(-x)|)dx < 4\delta r.$$

We now sketch how Lemmas 3.1 and 3.2 imply Theorem 1.1 for sigmoid units.

*Sketch of proof of Theorem 1.1.* The sigmoid function $\sigma$ with sharpness $s$ is not even in $L^1(\mathbb{R})$, so it is unsuitable as the function $\psi$ of Lemma 3.2. Instead, we define $\psi$ to be an affine combination of $\sigma$ gates as in Eq. (3.1). Then $\psi$ satisfies the hypotheses of Lemma 3.2.

Let $\theta = 4r$ and let $\bar{\gamma} = O(\theta^2/n)$ be as given by the statement of Lemma 3.1. Let $\epsilon = \bar{\gamma}/(2\lambda)$, and let $M = O(\sqrt{n}\log(n/(\epsilon\theta)))$ and $\delta = \Omega(\epsilon^3\theta/\sqrt{n})$ be as given by the statement of Lemma 3.1. By Lemma 3.2, there is $m \in \mathbb{N}$ and functions $h : \mathbb{R}^m \to \mathbb{R}$ and $g : \mathbb{R} \to \mathbb{R}^m$ such that $\phi = h \circ \psi \circ g$ is $(M, \delta, \theta)$-quasiperiodic and satisfies the hypotheses of Lemma 3.1. Therefore, we have a family $\mathcal{C}_0$ of affine functions $f : \mathbb{R}^n \to \mathbb{R}$ such that for $\mathcal{C} = \{\phi \circ f : f \in \mathcal{C}_0\}$ satisfies $\epsilon\text{-SDA}(\mathcal{C}, D, \bar{\gamma}) \geq 2^{\Omega(n)}\epsilon\theta^2$. Therefore, the functions in $\mathcal{C}$ satisfy the hypothesis of Theorem 2.1, giving the query complexity lower bound.

All details are given in the full version of the paper [18]. $\qquad\square$

## 3.1 Different activation functions

Similar proofs give corresponding lower bounds for activation functions other than sigmoids. In every case, we reduce to gates satisfying the hypotheses of Lemma 3.2 by constructing an appropriate $L^1$-function $\psi$ as an affine combination of of the activation functions.

For example, let $\sigma(x) = \sigma_s(x) = \max\{0, sx\}$ denote the ReLU unit with slope $s$. Then the affine combination

$$\psi(x) = \sigma(x + 1/s) - \sigma(x) + \sigma(-x + 1/s) - \sigma(-x) - 1 \qquad (3.2)$$

is in $L^1(\mathbb{R})$, and is zero for $|x| \geq 1/s$ (and hence has the mean bound property and essential radius $O(1/s)$). The proof of Theorem 1.1 therefore goes through almost identically, the slope-$s$ ReLU units replacing the $s$-sharp sigmoid units. In particular, there is a family of single hidden layer NNs using $O(s\sqrt{n}\log(\lambda sn))$ slope-$s$ ReLU units, which is not learned by any SQ algorithm using fewer than $2^{\Omega(n)}/(\lambda s^2)$ queries to $\mathrm{VSTAT}(O(s^2 n))$, when inputs are drawn i.i.d. from a logconcave distribution.

Similarly, we can consider the *s-sharp softplus* function $\sigma(x) = \log(\exp(sx) + 1)$. Then Eq. (3.2) again gives an appropriate $L^1(\mathbb{R})$ function to which we can apply Lemma 3.2 and therefore follow the proof of Theorem 1.1. For softsign functions $\sigma(x) = x/(|x| + 1)$, we use the affine combination

$$\psi(x) = \sigma(x + 1) + \sigma(-x + 1).$$

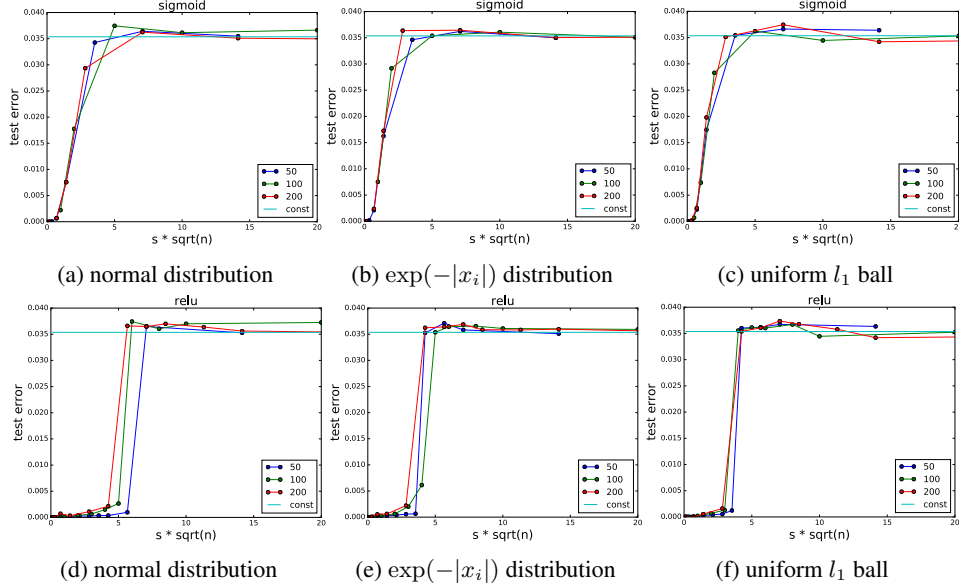

|     |     |     |
| --- | --- | --- |
| (a) normal distribution | (b) $\exp(-|x_i|)$ distribution | (c) uniform $l_1$ ball |
| (d) normal distribution | (e) $\exp(-|x_i|)$ distribution | (f) uniform $l_1$ ball |

Figure 4.1: Test error vs sharpness times square-root of dimension. Each curve corresponds to a different input dimension $n$. The flat line corresponds to the best error by a constant function.

In the case of softsign functions, this function $\psi$ converges much more slowly to zero as $|x| \to \infty$ compared to sigmoid units. Hence, in order to obtain an adequate quasiperiodic function as an affine combination of $\psi$-units, a much larger number of $\psi$-units is needed: the bound on the number $m$ of units in this case is polynomial in the Lipschitz parameter $\lambda$ of the query functions, and a larger polynomial in the input dimension $n$. The case of other commonly used activation functions, such as ELU (exponential linear) or LReLU (Leaky ReLU), is similar to those discussed above.

## 4 Experiments

In the experiments, we show how the errors, $\mathbb{E}(f(x) - y)^2$, change with respect to the sharpness parameter $s$ and the input dimension $n$ for two input distributions: 1) multivariate normal distribution, 2) coordinate-wise independent $\exp(-|x_i|)$, and 3) uniform in the $l_1$ ball $\{x : \sum_i |x_i| \le n\}$.

For a given sharpness parameter $s \in \{0.01, 0.02, 0.05, 0.1, 0.2, 0.5, 1, 2\}$, input dimension $d \in \{50, 100, 200\}$ and input distribution, we generate the true function according to Eqn. 1.1. There are a total of 50,000 training data points and 1000 test data points. We then learn the true function with fully-connected neural networks of both ReLU and sigmoid activation functions. The best test error is reported among the following different hyper-parameters.

The number of hidden layers we used is 1, 2, and 4. The number of hidden units per layer varies from $4n$ to $8n$. The training is carried out using SGD with 0.9 momentum, and we enumerate learning rates from 0.1, 0.01 and 0.001 and batch sizes from 64, 128 and 256.

From Theorem 1.1, learning such functions should become difficult as $s\sqrt{n}$ increases over a threshold. In Figure 4.1, we illustrate this phenomenon. Each curve corresponds to a particular input dimension $n$ and each point in the curve corresponds to a particular smoothness parameter $s$. The x-axis is $s\sqrt{n}$ and the y-axis denotes the test errors. We can see that at roughly $s\sqrt{n} = 5$, the problem becomes hard even empirically.

### Acknowledgments

The authors are grateful to Vitaly Feldman for discussions about statistical query lower bounds, and for suggestions that simplified the presentation of our results, and also to Adam Kalai for an inspiring discussion. This research was supported in part by NSF grants CCF-1563838 and CCF-1717349.

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
