[Reviews · NeurIPS 2017]

Reviewer 1



The authors proved a lower bound suggests that even the combination of small network size, smooth, standard activation functions, and benign input distributions is insufficient to make learning neural networks easy, which is quite interesting. However, we expect the authors provide more discussions on how to avoid such poor situations actively.

Reviewer 2



The paper gives an exponential lower bound to the complexity of learning neural networks in the statistical query model. The result is significant in that it applies to a large class of natural learning algorithms (such as SGD) and its assumptions are also natural. The proof is based on extending the statistical dimension approach to the present framework. The paper is a technically solid, elegant piece of work which brings together several research directions. One small comment: the connection to [12] described in Section 1.3 could be more specific.

Reviewer 3



The paper presents a new information-theoretic lower bound for learning neural networks. In particular, it gives an exponential statistical query lower bound that applies even to neural nets with only one hidden layer using common activation functions and log-concave input distributions. (In fact, the results apply to a slightly more general family of data distributions.) By assuming log-concavity for the input distribution, the paper proves a lower bound for a more realistic setting than previous works with applied to discrete distributions. To prove this SQ lower bound, the paper extends the notion of statistical dimension to regression problems and proves that a family of functions which can be represented by neural nets with one hidden layer has exponentially high statistical dimension with respect to any log-concave distribution. (I was surprised to read that statistical dimension and SQ lower bounds have not been studied for regression problems before; however, my cursory search did not turn up any previous papers that would have studied this problem.) It is not clear to me if there are any novel technical ideas in the proofs, but the idea of studying the SQ complexity of neural networks in order to obtain lower bounds for a more realistic class of nets is (as far as I can tell) novel and clever. The specific class of functions which are used to prove the lower bound is, like in most lower bound proofs, pretty artificial and different from functions one would encounter in practice. Therefore the most interesting consequence of the results is that reasonable-looking assumptions are not sufficient to prove upper bounds for learning NN's (as the authors explain in the abstract). On the other hand, the class of functions studied is not of practical interests, which makes me question the value of the experiments in the paper (beyond pleasing potential reviewers who dislike papers without experimental results). With the tight page limits of NIPS, it seems like the space taken up by the experimental results could have a better use. My main problem with the paper is the clarity of the write-up. I find the paper hard to read. I expect technical lemmas to be difficult to follow, but here even the main theorem statements are hard to parse with a lot of notation and formulas and little intuitive explanation. In fact, even the "Proof ideas" section (1.4), which I expected to contain a reader-friendly, high-level intuitive explanation of the main ideas, was disappointingly difficult to follow. I had to reread some sentences several times before I understood them. Of course, it is possible that this is due to my limitations instead of the paper's, but the main conceptual ideas of the paper don't seem so complex that this complexity would make it impossible to give an easily readable overview of the theorems and proofs. Overall, this seems to be a decent paper that many in the NIPS community would find interesting to read but in my opinion a substantial portion of it would have to be rewritten to improve its clarity.